# Environmental Practices That Have Positive Impacts on Social Performance: An Empirical Study of Malaysian Firms

**Noorliza Karia ***[ID] **and Ruben Charles Davadas Michael**

Operations Management, Universiti Sains Malaysia, Minden 11800, Penang, Malaysia; rayruben7@yahoo.com
* Correspondence: noorliza@usm.my

**Abstract:** Despite many environmental studies, the literature lacks studies emphasizing the significant nature of the human–environment connection. This study focuses on the impact of manufacturing environmental practices on social performance, which is crucial for employees' wellbeing, human development, and quality of life that lacks empirical evidence. This study searches for a mechanism to enhance social performance through sustainable practices and test the mediating effects of environmental collaboration. This study examines the proposed hypotheses on the data sample of 120 Malaysian manufacturing firms, with partial least squares structural equation modeling. Explicitly, the results reveal sustainable practices comprised of purchasing social responsibility (PSR), long term orientation (LTO), supplier assessment (SA), and environmental collaboration (EC) contribute almost 50% of social performance. Still, LTO and SA are the best practices. PSR, LTO, and SA significantly contribute 45% of EC, but LTO and PSR remain the best sustainable practices. Firms investing in these sustainable practices of improving social performance, driven by sustainability, show these are worthy strategies. Concentrating on certain sustainable practices could improve employees' wellbeing, human development, and quality of life. The novel contribution of the study is the formulation of social performance and its empirical work testing the mediating effects of EC between sustainable practices and social performance.

**Keywords:** sustainable practices; social performance; environmental collaboration; purchasing social responsibility; long term orientation; supplier assessment; human–environment connection

## 1. Introduction

Manufacturing sectors are responsible for 30% of carbon dioxide emissions, requiring an urgent commitment to manufacturing environmental practices [1]. Firms' persistence in sustainability meets the environmental demand of stakeholders (e.g., consumers, employees, communities, suppliers, distributors, and shareholders) to improve the negative impact of firms' activities on the planet and people [2–4]. Research shows that sustainable practices positively impact environmental performance and social performance [4] and, in turn, positively affect a firm's economic performance [5,6]. In contrast, poor sustainable practices, such as suppliers' unethical conduct and corrupt behavior, the low-quality level of product and poor working conditions, will lead to negative impacts on corporate image and reputation, the costs of recalls, environmental cleaning, government penalties and operations performance in the long term [6].

In a dynamic environment, the future competitiveness of manufacturing firms leans on employees who are responsible, directly or indirectly, for environmental sustainability. Employees play a significant role in executing sustainable practices, enabling and improving ecological sustainability, leading to profit and competitive advantage, and the firm's image and reputation and society's health and safety [7–11]. Besides positive effects on green products, the planet, and profit, environmental manufacturing practices also enhance people's wellbeing, human development, and quality of life [9,10]. Literature acknowledges the positive relationship between sustainable manufacturing practices and

social performance [12,13]. Still, the direction is unclear and limited studies identify which manufacturing environmental practices positively impact social performance [9,14]. This study addresses these gaps by identifying and examining the effects of sustainable practices on social performance.

Consequently, from stakeholders' concerns of environmental degradation has emerged the concept of environmental practices for maximizing benefits for people, about which there is considerably little in the extant literature of sustainability [15–17]. Though the people aspects of sustainability are crucial for both social and environmental performance improvement, there is a lack of empirical evidence on the relationship between sustainable practices and social performance [12,18,19]. Hence, unable to generalize the findings as the social implications of sustainable practices stay vague, no conclusive and consistent results have emerged [20]. Many previous studies maintain focus on the effects of sustainable practices on operational, economic, and environmental performance [21–26], neglecting social performance [16,27,28]. Others found a mixed relationship between sustainable practices and social performance [12,16,29]. Based on the natural resource based view (NRBV) theory, this study extends research on environmental sustainability and social performance thoughtfully in a significant area deserted by researchers [1,30,31].

Despite many environmental studies, literature has limited observations on the human–environment connection [32]. Therefore, evidence is lacking concerning the direction, mediation, or moderation effects of the sustainable practices–social performance relationship. The benefits of adopting sustainable practices can positively impact socioenvironmental performance [29,32]. However, Miemczyk and Luzzini's [20] research discovers that only certain environmental practices can positively impact social performance. In contrast, sustainable practices have an insignificant effect on social performance but can improve environmental performance, indirectly increasing financial and social performance [4]. Hence, no study has unearthed the black box of these inconclusive findings on the human–environment connection [4,29,32,33].

Enthusiastically highlighting such gaps, this study provides the novel contribution of sustainable practices–social performance in the manufacturing industry. Besides studying interjections required but restricted to research fields, it also advances thoughts for developing sustainability in manufacturing firms and the alternative environmental practices for cleaner production firms. Moreover, this study provides the first empirical work that examine the extent and impact of environmental concerns built in to certain sustainable practices, and tests the mediation effects of environmental collaboration in the sustainable practices–social performance relationship. This study is essential for those firms that aim to implement and invest in certain environmental practices. The empirical evidence helps managers, employees, and organizations better understand the importance of green, sustainable or ecological practices and their impact on social performance. Firms should be aware of the human–environmental connection, since humans cause most environmental degradation. Thus, firms need to integrate ecological practices to manage environmental, economic, and social performance in their strategic planning. Therefore, the findings contribute new work models that foster environmental practices and business sustainability in general and social aspects. The study results add to the literature about sustainability, human resources, social performance, and manufacturing sectors.

This paper is organized as follows. The following section presents the theoretical framework and hypotheses of the study, followed by methodology and analyses used, and results. The discussion section includes study findings and implications and concludes with limitations and suggestions for further research in the Conclusions section.

## 2. Theoretical Framework and Hypotheses

Future sustainable business depends on human capital as a booster of a firm's concerns regarding preserving the natural environment, sequentially increasing green products/services, and improving economic and social performance. Resource based view (RBV) theory endorses that such human capital has great potential to generate firm value

and sustainable competitiveness; hence, firms' responsibility for people is vital to ascertaining long term success [30,34]. Therefore, organizations need to connect firms and employees engaging in environmental sustainability through sustainable practices [35].

Sustainable practice is relevant as cleaner productions offer firm capabilities, and create costly and durable resources while simultaneously creating sustainable competitiveness, thus exploiting the extension of RBV to the natural resource based view (NRBV) theory [36]. By integrating the natural environment into RBV, NRBV endorses environmental or sustainable practices that can generate operational or organizational capabilities [36,37] to enhancing the four Ps: product/service innovation, people—wellbeing, profitability, and planet preservation [38]. NRBV theory explains that cleaner production or environmental practices are determinants of environmental performance and, subsequently, a firm's profitability and competitiveness [9,14]. The theoretical foundation of NRBV shows evidence that the concept of sustainable practices is an essential effort in empowering an improvement in humankind's social benefits and the wellbeing of humanity, thus enhancing a strength between ecological and economic impact [9].

The literature lacks studies emphasizing the significant nature of the human–environment connection [7,10]. Lately, research into social benefits or social concerns and priorities have attracted attention [4,9,16,18]. Given the little exposure to the accessibility of human–environment research [3,12,39], this significant study is desirable to unearth these relationships. Thus far, social performance has not been extensively investigated [40], unlike financial, environmental performance, or the three dimensions simultaneously [18,29,39]. This contemporary study highlights literature on sustainability and manufacturing environmental practices that positively impact people using NRBV theory.

## 2.1. Conceptualization

Sustainable practices are about creating activities, intentions, concerns, initiatives, or a culture of sustainability/environmental protection that have positive impacts on the planet and its ecosystem [14]. A firm's initial efforts concerning social priorities are to adopt sustainable practices, aiming to obtain and enhance social performance. According to NRBV theory, sustainability/ecological practices constructed to preserve and sustain the natural environment for the future lead to benefits in many aspects [13,21]. For social aspects and priorities, the concept of sustainable practices is the creation of environmental intentions that have positive impacts on people and social factors such as quality of life, and the benefits of social safety for all stakeholders and the universe [4,9,38].

Meanwhile, social performance is the consequence of sustainable practices. Social performance attributes measure quality of life, employees' health and safety, employees' skills, the motivation and productivity of employees, employees' wellbeing and welfare, community health and safety, the job satisfaction levels of employees, equal treatment, human rights, safe and humane working conditions, reduction in absenteeism, enhancing image and reputation and customer satisfaction through lower prices [4,6,10,12,39–42].

NRBV theory claims sustainable practices as part of RBV theory, which generates the organizational capabilities of human capital to enhance the firm's growth [34,38] (Barney, 1991; Karia, 2020). However, studies on sustainability effects show inconsistent results due to limited empirical evidence. Theoretically, the argument regarding the social benefits of sustainable practices remain little observed. In practice, being socially responsible, such as establishing employee welfare programs, charity and environmentally friendly policies, leads to additional costs. Therefore, organizations have yet to see the benefits of adopting sustainable practices on social performance. According to Miemczyk and Luzzini [20], certain sustainable practices might have the great potential to have the most significant effect on social performance by engaging cooperation between buyers and supplier partners [22,43]. Consequently, there is a need to identify and investigate sustainable practices–social performance relationships.

The concept of sustainable practices is essential to empowering an improvement in the social benefits and wellbeing of humankind [9,44]. Thus, implementing sustainable practices

can improve employees' quality of life, motivation, and productivity and enhance company image, reputation, and customer satisfaction [45,46], and social performance [3,7,12].

Stakeholder theory endorses that firms incorporate an employee, supplier, customer, competitor, government, community, and media for stimulating positive organizational behavior to maximize benefits for people [3,4,40,41]. Firms strategically and proactively integrate social elements of sustainability and witness their social performance. According to Baah et al. [2], regulatory stakeholder pressures significantly influence social performance (e.g., social reputation and corporate social responsibility). However, not all sustainable practices lead to sustainability [9]. According to Miemczyk and Luzzini [20], social factors significantly intervene in the relationship between social priorities and social performance. The extent of a firm's sustainability efforts determines the enhanced social elements of sustainable practices.

Literature highlights certain sustainable practices that have a positive effect on social performance, such as firms' sustainable manufacturing process and sustainable supply chain [39], green purchasing and sustainable packaging [47], supplier assessment, and collaboration with suppliers [6,16,48], long term orientation (LTO) [49], supply chain management practices [12] and environmental performance [4]. Wang and Dai [29] found that internal socially responsible management practices positively affect firm social performance. Internal and external factors can encourage sustainability; however, internal factors are the most significant [29]. Conversely, green logistics practices [4], PSR [50], supplier assessment [16,29], and collaboration [29] have an insignificant impact on social performance. Sustainable practices can influence sustainability performance [42], requiring strategic collaboration between buyers and supplier partners [22,43]. However, it takes years to develop a good environmental collaboration relationship, subsequently exposing firms to imitations by rivals in order to realize competitiveness [51]. By establishing partnerships, firms and supply chain partners construct valuable organizational resource capability, to enhance sustainability performance [52].

## 2.2. Hypotheses Development

Firms implementing environmental sustainability should have an impact on social sustainability [1]. Thus, sustainable practices could be resource capabilities for firms that can positively affect social performance. Figure 1 shows the study hypotheses.

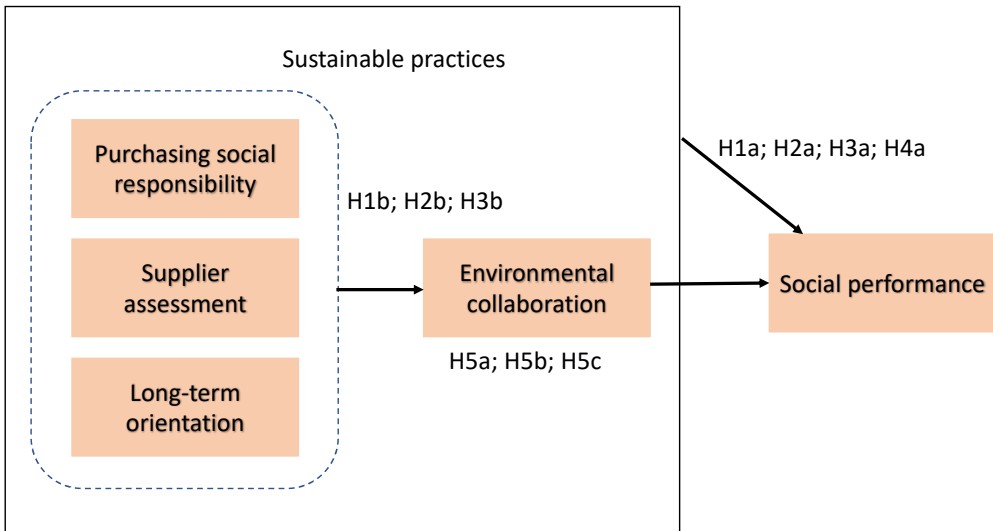

**Figure 1.** Research hypotheses.

Environmental purchasing is a critical sustainability issue for buyers and suppliers to consider [47,53]. Therefore, the concept of purchasing social responsibility (PSR) incorporates any socially responsible behavior with a purchasing function, socially responsible

purchasing firms, or purchasing managers' involvement in socially accountable purchasing activities [54]. Prendergast and Tsang [55] report that engaging in social responsibility consumption can significantly enhance social performance. Sustainable practices positively determine social performance [12]. Zailani et al. [47] found that environmental purchasing has a positive effect on social performance. As NRBV predicted, the focus on PSR should positively correlate with social performance, which requires strategic collaboration between buyer and supplier partners [22,43]. Therefore, the study's hypotheses are:

**Hypothesis H1a.** *PSR positively correlates to social performance.*

**Hypothesis H1b.** *PSR positively correlates to environmental collaboration.*

A sustainable relationship in terms of long term orientation (LTO) between the buyer and supplier is a significant factor for sustainable practices that can increase relationships, relational behavior, and satisfaction [40]. The buyer–supplier relationship will have a long term partnership/alliance or will continue for a long time as the rewards of the relationship continue [49]. Suppliers bound to buyer firms are more likely to effectively enhance social performance [56]. Relationship strength is too valuable for all partners' relationships; hence, supply chain partner stability is required [57,58]. Thus, LTO facilitates a firm's committed relationship, relational behavior, trust, and satisfaction, and are more likely to positively impact social performances [49,56]. NRBV suggests LTO, as an organizational capability, should positively affect social performance and collaboration. Therefore, the study hypotheses are:

**Hypothesis H2a.** *LTO positively correlates to social performance.*

**Hypothesis H2b.** *LTO positively correlates to environmental collaboration.*

Supplier assessment is an evaluation of the social performance of suppliers for suppliers to meet the social standard. It allows a firm to monitor, evaluate and select suppliers with a code of conduct, environmental or social criteria, to control sustainability related risk across supply chains [59]. Suppliers' appraisal and checking can reduce the risk of unethical conduct by assessing the actual examination and power of the suppliers to advance social performance or social concerns (e.g., working conditions, compliance with human rights) [29]. It subsequently enhances their interest in the environment and pressures them to consider social issues in supply chains [60]. Therefore, supplier assessment of sustainable practices provokes outstanding social performance by empowering collaboration within buyer and supplier firms' relationships. Sancha et al. [11] indicate that supplier assessment positively correlates with the buyer's social performance. Ni and Sun [59] confirm that supplier assessment leads to better environmental and social performance. NRBV suggests that supplier assessment as an operational capability should positively affect collaboration and social performance. Therefore, the study hypotheses are:

**Hypothesis H3a.** *Supplier assessment positively correlates to social performance.*

**Hypothesis H3b.** *Supplier assessment positively correlates to environmental collaboration.*

Environmental collaboration within buyer and supplier relationships requires maximum participation from all supply chain partners, requiring significant interaction, commitment, and involvement in the socially sustainable supplier practices [29]. Joint efforts amongst buying firms and their suppliers encourage amendments in supplier operations and activities to meet the social and ecological requirements [48,61] or partnership to empower trust and improve social issues and firms' social reputation. Collaboration positively correlates with social performance [11,16,59,62]. NRBV claims environmental collaboration as a manufacturing capability [37] that should positively affect social performance. Therefore, the study hypothesis is:

**Hypothesis H4a.** *Environmental collaboration positively correlates to social performance.*

The Role of Environmental Collaboration

While collaboration has been acknowledged as the essential determinant of a co-operative buyer–supplier relationship for sustainable practices and performance, most studies have two knowledge gaps. First, little research focuses on the collaboration–social performance relationship [29,63], restricting our understanding of theoretical driven and empirically proven explanations. Previous studies have inconsistent findings concerning the relationship between collaboration and social performance.

Recently, a study by Arora et al. [33] has proposed environmental collaboration as a mediator between sustainable strategic purchasing and organizational sustainability performance. Collaboration positively affects social and environmental performance [20,29] or better economic and social performance [64,65], but not necessarily better environmental performance [63]. In contrast, Wang and Dai [29] and Agyabeng-Mensah et al. [4] found that collaboration does not influence social performance. Therefore, the study proposes collaboration as a mediator to enhance the interaction between sustainable practices and social performance in determining the strong impact of sustainable practices on social performance.

To this end, the study extends the conventional framework in which sustainable practices are the primary mechanism to affect sustainability performance by anticipating that sustainable practices may have an intermediary impact on environmental collaboration before determining firm social performance [66]. The successful implementation of sustainable practices positively affects social performance and is more likely to increase the high levels of social performance through environmental collaboration. Therefore, the study hypotheses are:

**Hypothesis H5a.** *Environmental collaboration mediates the association of PSR and social performance.*

**Hypothesis H5b.** *Environmental collaboration mediates the association of long term orientation and social performance.*

**Hypothesis H5c.** *Environmental collaboration mediates the association between supplier assessment and social performance.*

### 3. Methodology

*3.1. Sample and Data Collection*

The research applies a survey to maximize the results' generalizability. It represents the population by gathering information from a large sample size that is very accurate because of the instrument designed [67]. However, this approach is a non-experimental study and is limited to a cross-sectional study by collecting data at a specific time. This study focuses on Malaysian manufacturers listed in the reliable directory http://www.matrade.gov.my (MATRADE, accessed on 3 January 2019) because they are the primary driver of the national economy. These firms also have the environmental standard 14,001 and implement various sustainable practices for preserving the environment, hence they are the right respondents to answer the survey questions. The study randomly selected 400 firms from the directory and distributed questionnaires through an email containing an online survey link by Google, which is more convenient for the respondent. It took from May to September 2019 to update and conclude for analysis. The respondents are senior managers and above who have extensive knowledge and experience in the concepts relevant to this study.

The study designed a questionnaire with simple instructions to increase the response rate from respondents and minimize measurement error [68], and adopted multi-item measurement scales to provide a robust measure of variables to reduce measurement error [69]. The cover letter included the study objective, consent to research ethics, and deadline. Questions were set as required to avoid answers with missing data within four weeks—three follow-up reminders were sent a week after initial emailing. Finally, 120 firms returned completed and useable questionnaires, which gives an overall 69% response rate.

The minimum sample size required for PLS model should be ten times larger than the number of variables in the study [70]. This study has five variables, hence, 50 samples is the minimum sample size, which establishes that 120 observations is good enough for the study to use SEM for the data analysis.

This study used Smart PLS software to perform SEM-PLS because it is an advanced multivariate statistical analysis technique that can simultaneously perform all types of analysis, including measurement and structural models, without losing the original data [71]. The relationships between exogenous and endogenous variables are tested using the structural equation modelling partial least square (SEM-PLS) technique. This study used SPSS for preliminary screening to make sure there was no missing value or any outliers. Then demographic data were generated using SPSS. Afterwards, this study imported the data into PLS to measure the structural and measurement models. Finally, this study tested all the hypotheses using path linkages of the model.

### 3.2. Instrument

All the measurement items were adopted from previous literature and then adapted into the study context. The study survey instrument includes twenty items of measurement constructs for four sustainable practices and four items of social performance measured by the following:

- Purchasing social responsibility: PSR included six items [72];
- Supplier assessment: SA included five items [6];
- Long term orientation: LTO included four items [49];
- Environmental collaboration: EC included five items [6];
- Social performance: SP included four items [6].

The study utilized the Likert scale pointing from "strongly disagree (1)" to "strongly agree (5)". Based on the pilot test participants (five and two experts, respectively, from the industry and academia), the study further reviewed and adapted the questionnaire to improve constructs and items used for the actual survey. The study variables were pilot tested on 15 firms that were not included in the sample. All the study variables have Cronbach's alpha values above 0.70, indicating the reliability of the items.

### 3.3. Sample Profile

Table 1 describes the firm profile of the respondents. The descriptive statistics show that slightly more than half (56.7%) of respondents have been in firms with corporate social responsibility for more than four years. The respondents are mainly from electrics and electronic industry (34.2%), followed by food industry (17.5%) and the automobile industry (14.2%), and almost equally representative for other industries. The firms are practically similarly represented in firm size; 28% have less than 251 employees, 33% have 251 to 500 employees, and 39% have more than 500 employees. Slightly more than half (57.5%) of the firms have quality supplier selection, followed by environmentally friendly (23.3%).

Table 2 presents the respondent's profile that almost equals representations of demographic variables. There is nearly the same percentage of gender; more than half of the respondents' age is between 25–35 years of age (61%). Slightly more than half of them hold a bachelor degree (56.7%), executives (57.5%) or managers (42.5%); and have been working less than five years (54.2%) or working five years and more (45.8%).

Overall, the data set showed no significant differences between groups ($p < 0.005$), suggesting that the collected data is free from response bias [73]. The scatter plot of the data set demonstrated randomly presenting normality, linearity and homoscedasticity. This study data is also free from common method variance bias, as the total explained variance is less than 50 per cent, indicating it is sufficient for inferential analysis. Table 3 presents the result of Harman's single factor test.

**Table 1.** Organization profile.

| Demographic | Categories | Frequency | Per Cent |
|---|---|---|---|
| Corporate Social Responsibility | <1 Year | 18 | 15 |
| | 1–3 years | 34 | 28.3 |
| | 4–6 Years | 24 | 20 |
| | >6 Years | 44 | 36.7 |
| Business | Electrics and Electronics Industry | 41 | 34.2 |
| | Textiles and Apparel Industry | 13 | 10.8 |
| | Medical Device Industry | 15 | 12.5 |
| | Food Industry | 21 | 17.5 |
| | Pharmaceutical Industry | 13 | 10.8 |
| | Automobile Industry | 17 | 14.2 |
| No. of Employees | 10–250 employees | 34 | 28.3 |
| | 250–500 employees | 39 | 32.5 |
| | >500 employees | 47 | 39.2 |
| Supplier Selection Criteria | Quality | 69 | 57.5 |
| | Innovation | 12 | 10 |
| | Environmental Friendly | 28 | 23.3 |
| | Price | 10 | 8.3 |
| | Others | 1 | 0.8 |

**Table 2.** Respondent's profiles.

| Demographic | Categories | Overall | | |
|---|---|---|---|---|
| | | Frequency | Per Cent | Cumulative Percent |
| Gender | Male | 56 | 46.7 | 46.7 |
| | Female | 64 | 53.3 | 100 |
| Age | <25 years | 14 | 11.7 | 11.7 |
| | 25–35 years | 73 | 60.8 | 72.5 |
| | 36–50 years | 31 | 25.8 | 98.3 |
| | 51–65 years | 2 | 1.7 | 100 |
| Educational Level | Secondary School | 2 | 1.7 | 1.7 |
| | Certificate/Diploma | 31 | 25.8 | 27.5 |
| | Bachelor's Degree | 68 | 56.7 | 84.2 |
| | Postgraduate | 19 | 15.8 | 100 |
| Current Job Position | General Manager | 8 | 6.7 | 6.7 |
| | Human Resource Manager | 10 | 8.3 | 15 |
| | Warehouse Manager | 13 | 10.8 | 25.8 |
| | Safety and Health Manager | 5 | 4.2 | 30 |
| | Supply Chain Manager | 15 | 12.5 | 42.5 |
| | Executive | 69 | 57.5 | 100 |
| Year of working experience | <5 years | 65 | 54.2 | 54.2 |
| | 5–10 years | 39 | 32.5 | 86.7 |
| | >10 years | 16 | 13.3 | 100 |

**Table 3.** Total variance explained.

| Component | Initial Eigenvalues | | | Extraction Sums of Squared Loadings | | |
|---|---|---|---|---|---|---|
| | Total | % of Variance | Cumulative % | Total | % of Variance | Cumulative % |
| 1 | 9.289 | 38.702 | 38.702 | 9.289 | 38.702 | 38.702 |

## 4. Result

### 4.1. Measurement Model

The study used the cut off point criteria suggested by [65] to assess the convergent validity of constructs. Table 4 shows the constructs and items for each construct, the primary/outer loadings, Cronbach's alpha values, composite reliability (CR), and average variance extracted (AVE). The convergent validity of the construct is satisfactory. The importance of all the constructs of AVE exceeded the recommended value of 0.5 [74]. Thus, this research model shows sufficient convergent validity. The CR values show more than 0.70, to justify further the model fit, as per another indicator proposed by [71].

**Table 4.** Convergent validity of measurement model.

| Constructs | Items | Loadings | Cronbach's Alpha | CR | AVE |
|---|---|---|---|---|---|
| SA | SA1 | 0.647 | 0.759 | 0.838 | 0.511 |
| | SA2 | 0.708 | | | |
| | SA3 | 0.807 | | | |
| | SA4 | 0.684 | | | |
| | SA5 | 0.717 | | | |
| EC | ECL1 | 0.728 | 0.842 | 0.888 | 0.614 |
| | ECL2 | 0.829 | | | |
| | ECL3 | 0.840 | | | |
| | ECL4 | 0.786 | | | |
| | ECL5 | 0.727 | | | |
| LTO | LTO1 | 0.702 | 0.774 | 0.853 | 0.594 |
| | LTO2 | 0.769 | | | |
| | LTO3 | 0.802 | | | |
| | LTO4 | 0.804 | | | |
| Responsibility | PSR1 | 0.715 | 0.812 | 0.870 | 0.572 |
| | PSR2 | 0.773 | | | |
| | PSR3 | 0.827 | | | |
| | PSR4 | 0.754 | | | |
| | PSR5 | 0.706 | | | |
| SP | SP1 | 0.848 | 0.785 | 0.861 | 0.608 |
| | SP2 | 0.752 | | | |
| | SP3 | 0.743 | | | |
| | SP4 | 0.771 | | | |

Additionally, all the values of Cronbach's alpha exceed the minimum criteria of 0.60. Finally, all the weights of factor loading were also more than 0.5, signifying that the constructs were fit. The results confirmed the convergent validity of the constructs.

Next, this study further verified that the model fit by using the discriminant validity test to ensure the constructs of the research model are unrelated. Table 5 presents the discriminant validity result that measures the correlation between constructs. The constructs have discriminant validity when the square root of AVE for each construct is more significant than other values lying vertically and horizontally [74]. In this study, the square

roots of AVE are higher than the correlation within and between constructs. In other words, the diagonal bolded values are all greater than their corresponding column and row values. Thus, this measurement model passed the discriminant validity test as well. The heterotrait–monotrait (HTMT) ratio was also used to investigate the discriminant validity [75]. The results confirmed the discriminant validity as the HTMT ratio of correlation values was less than 0.9. Finally, this model went through established criteria to verify the fitness or validity of the measurement model.

**Table 5.** Discriminant validity of measurement model.

|  | **SA** | **EC** | **LTO** | **PSR** | **SP** |
|---|---|---|---|---|---|
| SA | 0.715 |  |  |  |  |
| EC | 0.531 | 0.784 |  |  |  |
| LTO | 0.586 | 0.565 | **0.770** |  |  |
| PSR | 0.698 | 0.607 | 0.546 | **0.756** |  |
| SP | 0.589 | 0.525 | 0.596 | 0.580 | **0.780** |

*4.2. Structural Model*

The structural model is assessed after validating the measurement model by performing bootstrapping with 5000 resamples to obtain the R2 beta and the corresponding values [71]. The result of SEM and its graphical representation derived from Smart PLS software is also given in Figure 2.

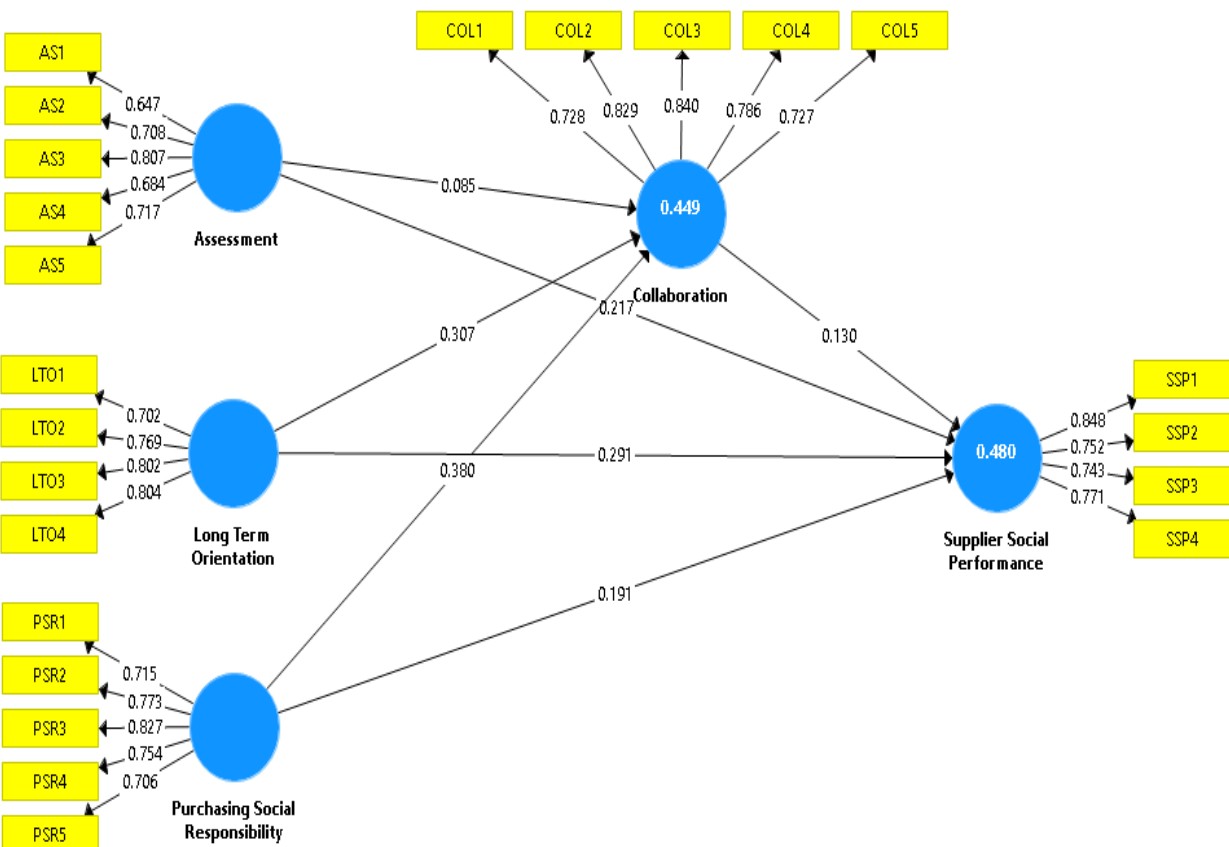

**Figure 2.** Research model.

Before assessing the structural relationships, collinearity must be examined to ensure it does not bias the regression results [71]. The VIF value is recommended if it is less than 5 [76]. As a result, no multicollinearity issue was found in the research model (Table 6). The values of R2 show that these independent variables can explain a percent of the variance in

the dependent variables (Table 7). The R2 is the combined effect of exogenous variables on endogenous variables; the model's predictive accuracy is moderate. R2, with 0.75, 0.50, and 0.25, respectively, describe substantial, medium, or weak levels of predictive accuracy [75]. This study measured the model's predictive relevance by applying the blindfolding procedure.

**Table 6.** VIF values of the structure model.

| Construct | VIF |
|---|---|
| SA | 2.210 |
| EC | 1.816 |
| LTO | 1.613 |
| PSR | 2.068 |
| SP | 2.223 |

**Table 7.** Coefficient of determination (R2).

| Construct | R Square | Effect |
|---|---|---|
| EC | 0.449 | Moderate |
| SP | 0.480 | Moderate |

The model has sufficient predictive relevance when the Q2 values are more than 0 (Table 8). The effect size on both social performance and collaboration is considered as having moderate predictive relevance. Hair et al. [71] describe the Q2 effect sizes at Q2 values higher than 0, 0.25, and 0.50 show, respectively, small, medium, and significant predictive relevance of the PLS path model.

**Table 8.** Predictive relevance.

| Construct | Value | Effect |
|---|---|---|
| EC | 0.262 | Medium |
| SP | 0.268 | Medium |

Structure model results (Table 9) showed that the sustainable practices could explain about 48% of the variance in SP and 45% of the variance in EC. LTO ($\beta$ = 0.291, t = 2.765, $p < 0.01$) and SA ($\beta$ = 0.217, t = 1.750, $p < 0.1$) have a positive significant relationship with SP, hence supporting H2a and H3a. While PSR ($\beta$ = 0.380, t = 3.633, $p < 0.000$) and LTO ($\beta$ = 0.307, t = 2.222, $p < 0.05$) have a positive significant relationship with EC, hence, supporting H1b and H2b.

**Table 9.** Structural model results.

| | Path | Beta | Standard Error | T Value | *p* Values | Decision |
|---|---|---|---|---|---|---|
| H1a | PSR -> SP | 0.191 | 0.131 | 1.456 | 0.146 | Not Supported |
| H1b | PSR -> EC | 0.380 | 0.105 | 3.633 | 0.000 | Supported |
| H2a | LTO -> SP | 0.291 | 0.105 | 2.765 | 0.006 | Supported |
| H2b | LTO -> EC | 0.307 | 0.138 | 2.222 | 0.026 | Supported |
| H3a | SA -> SP | 0.217 | 0.124 | 1.750 | 0.080 | Supported |
| H3b | SA -> EC | 0.085 | 0.163 | 0.523 | 0.601 | Not Supported |
| H4a | EC -> SP | 0.130 | 0.107 | 1.217 | 0.224 | Not Supported |

Purchasing social responsibility (PSR); long term orientation (LTO); supplier assessment (SA); environmental collaboration (EC); social performance (SP).

In contrast, PSR (t = 1.450) and EC (t = 1.217) have an insignificant relationship with social performance, while SA (t = 0.523) has an insignificant relationship with EC, hence, not supporting H1a, H4a, and H3b respectively.

The mediating effects are summarized in Table 10. The results showed that indirect effects. H5a (t = 1.06), H5b (t = 0.885) and H5c (t = 0.376), were not significant, while there was no mediating effect of EC, hence, H5a, H5b and H5c were not confirmed.

**Table 10.** Mediator analysis (indirect effect).

|  | Path | Beta | Standard Error | T Value | *p* Values | Decision |
|---|---|---|---|---|---|---|
| H5a | PSR -> EC -> SP | 0.049 | 0.047 | 1.06 | 0.289 | Not supported |
| H5b | LTO -> EC ->SP | 0.04 | 0.045 | 0.885 | 0.376 | Not supported |
| H5c | SA -> EC -> SP | 0.011 | 0.029 | 0.376 | 0.707 | Not supported |

## 5. Discussion and Implications

This study offers a viable model of social performance mechanisms through sustainable practices for improving environmental degradation. The study asserts that sustainable practices seriously determine social performance and environmental collaboration. Consequently, the study provides some novel findings denoting the antecedents and outcomes of sustainable practices. Sustainable practices are positively related to SP and EC, indicating that SP and EC are enhanced when sustainable practices increase. Sustainable practices, such as PSR, LTO, SA, and EC, are crucial to improve SP, whereby PSR, LTO, and SA enhance EC significantly. Manufacturing firms' commitment to LTO and SA are more effective in significantly influencing SP. At the same time, LTO and PSR are substantially more conducive to enhancing EC among Malaysian manufacturing firms, a result not affirmed quantitatively by previous studies [31]. These variables, used to test their impact on social performance, are novel, since they are almost non-existent in past research [15]. Past studies paid less attention to social aspects and failed to determine which practices affect social performance.

The findings confirm that LTO and SA, and LTO and PSR, are essential sustainable practices positively correlating to SP and EC, respectively, enabling and enhancing social sustainability. This empirical evidence shows that sustainable practices are the organizational capabilities of social performance and environmental collaboration, supporting NRBV theory. In line with existing studies, sustainable practices determine social performance significantly [7,12,32]; SA has a significant positive impact on social performance [11,59].

These findings underline that sustainable practices are unlikely to be implemented without strategic environmental collaboration and initiative efforts between buyers and suppliers partners. Hence, social understanding and ecological cooperation are enhanced when LTO, PSR, and SA practices increase. Remarkably, LTO is essential for empowering SP and EC improvement, which extends previous knowledge and research by integrating multidimensional constructs of sustainable practices focusing on social–environmental aspects and partnerships into the research model.

In contrast, the study results highlight that PSR and EC do not directly impact SP, consistent with the previous studies in China [29] and Ghana [4]. These findings challenged prior studies that PSR [47] and EC [11,16,72] positively affect social performance. Further, SA does not directly influence EC. Still, SA is more likely to affect the buyer–supplier relationship or continuous monitoring, strengthening the facilitation of collaboration [11]. However, SA and EC are necessary to complement each other to strengthen social safety and reputation. This supports Ni and Sun [59], who reported that the combined use of both leads to better environmental and social performance globally. Interestingly, EC did not mediate the sustainable practices–social performance relationship, suggesting that EC is the outcome of sustainable practices, not a mediator [33,60]. These imply that Malaysian manufacturing firms have promoted environmental collaboration, excellent social benefits, and human development through sustainable practices initiatives.

Notably, Malaysian manufacturing firms have executed a high degree of sustainable practices in PSR, supplier assessment, LTO, and environmental collaboration and witnessed high social performance and ecological cooperation. In addition, the advantages of implementing these sustainable practices enhance social performance and environmental

cooperation. Sustainable practices and Malaysian manufacturers' commitment and responsibility towards humanity improved green human resources and partnership. This denotes that firms' concerns for staff determine firms' growth and sustained competitiveness by executing sustainable practices as a booster of employee performance.

### 5.1. Theoretical Implications

NRBV theory denotes the understanding that sustainable practices create organizational capability and serve as a function of social performance and environmental collaboration. Based on NRBV theory, this study connects sustainable practices to positive impacts on employees' quality of life, safety, and wellbeing, and to affect social performance and environmental collaboration, subsequently enhancing environmental performance and sustained competitive advantage [30]. In sustainability literature, social sustainability is the relevant theoretical lens to make employees feel more positive, an area that has received limited attention [9,12,14,30]. Therefore, this novel study contributes to a viable mechanism of a sustainable practices model to promote manufacturing firms' social performance and environmental collaboration. Hence, the study addresses the gap by investigating sustainable practices–social performance relationships and provides theoretically driven and empirical evidence for the generalizability of the study. This study unearths the black box of sustainable practices' effects on social performance and environmental collaboration from a manufacturers' view rather than global and inconclusive findings.

The study identifies the emergent sustainable practices that influence social performance and promote environmental collaboration, which were frequently deserted by authors [16–19]. The study identifies, conceptualizes, and validates attributes of sustainable practice—PSR, LTO, SA, and EC—as predictors of social performance and PSR, LTO, and SA as outcomes of EC. The study also confirms LTO and SA have a positive correlation with social performance, and LTO and PSR have a positive correlation with environmental collaboration that has not been revealed before, they are, hence, novel contributions. Thus, these empirical findings validate the positive impact of sustainable practices on social performance and environmental collaboration quantitatively [31] and explain the mixed results of previous studies [12,18–20].

### 5.2. Managerial Implications

The findings provide several directions for firms, stakeholders, and managers to encourage social performance and environmental collaboration, and, subsequently, sustainable competitiveness, by preserving the natural environment. Firms can add value to the nobility of humans by maximizing the benefits of employees and supporting human capital through sustainability. Firms switching corporate social responsibility towards environmental sustainability through cleaner production enable sustainable practices to positively impact social sustainability, profitability and competitiveness. These findings help firms and managers understand the effects of sustainable practices on social performance and environmental collaboration, improving to firms' image and reputation, profitability, and sustained competitiveness. Sustainable practices allow cooperation and strengthening buyer–supplier relationships' performance. Instead of investing in all sustainable practices, firms can invest in LTO and SA to attain high social sustainability while implementing LTO and PSR to enhance the significant effect of environmental collaboration.

### 6. Conclusions

Based upon NRV theory, our findings prove the critical role of sustainable practices in enhancing social performance and environmental collaboration. The study provides important implications for industry and practitioners to implement ecological sustainability for improving the four Ps of sustainability: product/service innovation, people wellbeing and development, profitability, and planet preservation.

Despite the novel contributions, this paper has its limitations. First, though the research discovers significant results and generalizations, the social performance model is



most durable for Malaysian manufacturing firms' committed to environmental practices. The model of social performance only captured some of the sustainable practices. Considerable knowledge and research are welcome as the social implications of sustainable practices become generalizable and well defined. The research model anticipated that sustainable practices could positively impact social performance and environmental collaboration, where PSR, LTO, SA, and EC are potential sustainable practices. The model proves the direct, indirect, and absent mediation effect of EC.

Consequently, much research should give more attention to the human–environment relationship. Future research should conduct a similar study in identical or different industries, firms, and countries for model validation. The following research model should predict additional sustainable practices, significant moderators, or mediators on social performance, economic and environmental, in similar or different industry and country contexts. Future studies should further analyze the influences between practices and the extent to which practices influenced sustainability performance directly or indirectly. Finally, future research can use different techniques and approaches to explore the model of a social account by conducting a longitudinal study for further model validation and generalization, as the outcomes may vary and change over time.

**Author Contributions:** Conceptualization, N.K.; Data curation, R.C.D.M.; Formal analysis, R.C.D.M.; Methodology, R.C.D.M.; Writing—original draft, N.K.; Writing—review & editing, N.K. All authors have read and agreed to the published version of the manuscript.

**Funding:** This research was funded by Research University Grant, Universiti Sains Malaysia, 1001/PMGT/8016031.

**Institutional Review Board Statement:** Not applicable.

**Informed Consent Statement:** Not applicable.

**Data Availability Statement:** Not applicable.

**Acknowledgments:** The authors would like to thank Universiti Sains Malaysia for funding this research under the Research University Grant (1001/PMGT/8016031).

**Conflicts of Interest:** The authors declare no conflict of interest.

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
