# Peer review of "Environmental Practices That Have Positive Impacts on Social Performance: An Empirical Study of Malaysian Firms"

_sustainability, doi:10.3390/su14074032_

Round 1

Reviewer 1 Report

The work " Environmental practices and social performance" sent for review is very important and necessary from an economic, social, and ecological point of view.

I have read the manuscript with interest and find that it broadly meets the criteria for publication. It is well-written and has an analytical and research character. However, there are some shortcomings that need to be reviewed. They are addressed in relation to the different sections of the article.

  1. First of all, I suggest clarifying the title of the manuscript. It is too general in its present form. The research concerned only 120 manufacturing companies from Malaysia. This fact should be clearly stated.
  2. In the Abstract, the authors write: "This study examines the proposed hypotheses on the data sample of 120 manufacturing firms, with partial least squares structural equation modelling." Is the sample of only 120 observations big enough to use SEM for data analysis?
  3. Who is the article addressed to? What are the benefits for the reader of the presented work? The answer to these questions should become part of the introduction.
  4. I suggest a wider description of how the sample is selected. In my opinion, it must be clearly stated whether the selection of the sample was random or non-random. Even with the most accurate methods of analyzing test results, the results obtained on non-random samples should be treated with caution.
  5. In order to collect data for analysis, the authors used an internet survey technique. Why did the Authors consider it to be suitable, despite its limits and disadvantages? The size of the measurement error should be added.
  6. The discussion of the results should be extended by adding the analysis of similar works created by other researchers.
  7. The authors do not mention the issue of the limitations of their research in any of the sections. They also do not indicate any further directions of research in this area. These aspects need to be completed.

I hope that the indicated remarks will help the Authors to improve their text so that the work will be published. Good luck!

Author Response

Dear Reviewer,

Thank you for your constructive comments to enhance the manuscript. I hope my responses satisfy your concerns.

Regards

K. Noorliza

Reviewer 2 Report

Dear Author/s,

Re: Manuscript “Environmental practices and social performance”

Reviewer’s report:

The paper deals with an interesting topic, current and scarcely treated, such as the connection of sustainable practices-social performance in the manufacturing industry, due to the importance of creating environmental intentions to have positive impacts on people. The article is well written, well structured, the results are interesting and adequately discussed with the theoretical framework. Also, the theoretical and managerial implications are relevant and the lines of research show signs of great research maturity.

However, it would be interesting to determine the importance of the case studied, justify the case of the manufacturing companies participating in the sample, specify the country of these participating companies or the date of data collection. It seems that the case study is that of Malaysian manufacturing companies, but nothing is explained about them and the importance of the case in point 3.3. Sampleprofile.

There are also some minor errors such as the fact that a period has not been added at the end (line 532) and some acknowledgments appear without content (line 533).

The bibliography should be updated. A simple search for the keywords of the title “Environmental practices” and “social performance” provides about 85 results between the years 2021 and 2022.

Best regards

Author Response

Dear reviewer,

Thank you for your brilliant comments. 

Regards,

K. Noorliza

Reviewer 3 Report

The article is of interest, well written, well structured. The research methodology is well argued, the results obtained are relevant and well explained. 

Author Response

Thank you for your kind compliment about the article. 

We really appreciate it.

Regards

liza

Reviewer 4 Report

Dear authors, thank you for the opportunity you gave me to read your manuscript. The paper refers to an important and actual problem.I found it interesting and hope that the comments that I have made prove to be constructive and help the authors further refine their manuscript.

The overview is clear and well presented. It should be emphasized that there was a significant return of responses from enterprises.

The methodology should be described in detail. Why did the authors use these methods? What are the advantages and limitations of these methods? According to the authors, can the industry in which the companies operate have an impact on the results of the study?

The results should be more discussed. Results and discussion should be more related to the past research. Please complete this point. I would more emphasize the practical use of research results. 

Author Response

Dear reviewer,

Thank for your brilliant comments

Regards,

K. Noorliza
